


# The assessment of earthquake-triggered landslides
# susceptibility with considering coseismic ground
# deformation
**Yu Zhao[1, 2], Zeng Huang[1], Zhenlei Wei[1*], Jun Zheng[1], Kazuo Konagai[3]**
1.  College of Civil Engineering and Architecture, Zhejiang University, Hangzhou 310058, China
2.  MOE Key Laboratory of Soft Soils and Geoenvironmental Engineering, Zhejiang University,
Hangzhou 310058, China
3.  International consortium on landslides, Kyoto 611-0011, Japan
* Contact author: Zhenlei Wei, email: weizhenlei@zju.edu.cn
## Abstract
The distance to the surface rupture zone has been commonly regarded as an important
influencing factor in the evaluation of earthquake-triggered landslides susceptibility.
However, the obvious surface rupture zones usually do not occur in some buried-fault
earthquakes cases, which mean lacking of the information about the distance to the
surface rupture. In this study, a new influencing factor named coseismic ground
deformation was added to remedy this shortcoming. The Mid-Niigata prefecture
earthquake was regareded as the study case. In order to select a more suitable model for
generating the landslides susceptibility map, three commonly used models named
Logistic Regression (LR), Artificial Neural Networks (ANN) and Support Vector
Machines (SVM) were also conducted to assess the landslides susceptibility. The
performances of these three models were evaluated with the receiver operating



characteristic (ROC) curve. The calculated results showed the ANN model has the
highest AUC (area under the curve) value of 0.82. As the earthquake triggered more
landslides in the epicenter area, which makes it more prone to landslides in further
earthquakes, the landslides susceptibility in the epicenter area was also further
evaluated.
**Keywords:** Earthquake triggered landslides; Landslide susceptibility mapping;
coseismic ground deformation;

## 1. Introduction

Earthquake-triggered landslides are commonly seen in the earthquake disaster chain.
The landslides not only bring loss of life and property but also seriously affect the post-
earthquake rescue. By summarizing the data of 40 historical earthquakes events in the
world, Keefer discovered that the earthquake-triggered landslide was the main reason
for the loss of life and property (Keefer, 1984). More than 60 people were killed and
nearly 100,000 people were displaced due to the Mid-Niigata earthquake in 2004
(Bandara and Ohtsuka, 2017). In 2008, the Wenchuan earthquake triggered nearly
200,000 landslides, killing about 20,000 people (Xu et al., 2012b). At present,
numerous researchers regarded the susceptibility mapping as an effective way to hazard
mitigation and disaster management, and a number of models have been used to
generate landslide susceptibility maps.
At present, one type of commonly used methods to evaluate the susceptibility of
landslides is the physical-based method. For this type of method, the study area is


usually divided into slopes units and then LEM or FEM are applied to calculate the
safety factor (FS) of each slope unit (Saade et al., 2016). However, the physical
mechanism of the landslide is often very complicated, especially for the landslides
caused by earthquakes. Due to the difficulty of obtaining enough parameters for slope
dynamic analysis, it still is a tough job to assess the landslide susceptibility with
physical-based models in large-scale areas.
The statistical learning method was the another important method for landslide
susceptibility assessment. This type of method is based on the assumption that future
landslides would be easily to occur under similar conditions to those of the previous
landslides. By analyzing the characteristics of the current landslides, a set of influencing
factors are usually selected to implement statistical learning and evaluate the landslide
susceptibility map (Pham et al., 2017; Ali et al., 2019; Lin et al., 2019). At present,
many statistical learning methods have been used successfully to calculate the
landslides susceptibility index (LSI) and generate the earthquake-triggered landslide
susceptibility maps (Hong et al., 2017; Pham et al., 2016; Xu et al., 2012a; Yi et al.,
2019). For example, Yang et al., (2015) established the susceptibility map of seismic
landslides for the Lushan earthquake in Sichuan Province with an artificial weighting
method. Shrestha and Kang (2019) used a maximum entropy model to produc the
landslide susceptibility map of the central region of the Nepal Himalaya. However, the
relatively good performance of these methods highly relies on the local geo-
environment factors and self-features of the methods. For different study areas, the most
accurate method is also different. Thus, it is necessary to make comparisons between





various methods for selecting a more suitable method which produces a more reliable
landslide susceptibility map (Bui et al., 2016).
Gorum et al. (2011) pointed out that the influencing factors of seismic landslide should
include seismic correlation parameters, geology parameters, and topography
parameters. Ding and Hu (2014) conducted the cluster analysis and the maximum
possible classification method to study seismic landslides susceptibility of Beichuan
County in the Wenchuan earthquake. Influencing factors contain land-use type, seismic
intensity, and annual rainfall were selected to produce a reasonable susceptibility map.
Since the earthquake-triggered landslides tend to occur frequently near the surface
rupture zone (Xu et al., 2012b; Xu, 2014). Numerous scholars took the distance to the
surface rupture zone as an influencing factor in the evaluation of landslides
susceptibility (Xu et al., 2012b; Xu, 2014). However, it is worth noting that some
buried- rupture earthquakes often do not have obvious surface rupture zones, the buried
rupture earthquakes can also trigger abundant landslides (Xu, 2014). The evaluation
accuracy of landslide susceptibility for buried rupture earthquake is affected by a lack
of the factor of the distance to rupture (Regmi et al., 2016). Therefore, it is necessary
to improve the accuracy of landslides susceptibility assessment for buried rupture
earthquakes by introducing new influencing factors.
The Mid-Niigata Earthquake, which occurred in 2004, has become an important case
for studying landslides due to good seismography and rich collection of seismic
landslides. Wang et al., (2007) detected the relationship between landslide occurrence
with geological, geomorphological conditions, slope geometry, and earthquake



parameters for the Mid-Niigata earthquake. Bandara and Ohtsuka (2017) used landslide
occurrence ratio (LOR) to determine the correlation between the occurrence of
earthquakes triggered landslides and geological attributes for the Mid-Niigata
earthquake.
In this paper, based on GIS technology, three statistical methods and two different scales
are evaluated to assess the landslides susceptibility caused by the Mid-Niigata
earthquake. First of all, we selected lithology, elevation, slope, slope aspect, surface
curvature, distance from the road and the peak value of earthquake acceleration as the
influencing factors to evaluates the susceptibility of seismic landslides in the whole
affected zone (large-scale area). For large-scale area, three different statistical learning
methods (logical regression (LR), Support Vector Machine (SVM), and artificial neural
network (ANN)) are utilized and compared to make reasonable seismic landslides
susceptibility map. As the epicenter area that has higher landslide frequency more prone
to earthquake-triggered landslides, the seismic landslides susceptibility in this area is
further evaluated. Finally, given the fact of very short surface ruptures, the Mid-Niigata
earthquake was regarded as a buried rupture earthquake (Maruyama et al., 2007). The
coseismic ground deformation decomposed from high-resolution DEM is added as an
influencing factor in order to improve the evaluation accuracy of the seismic landslide
susceptibility for the epicenter area.
**2.  Study area**
The Mid-Niigata earthquake occurred on October 23, 2004, The Japan Meteorological


Agency (JMA) measured the magnitude of the mainshock is 6.8, the epicenter is located
at 37°18'16. 56"N, 138°50'10. 32"E, the focal depth is about 13.1 km (Chigira and Yagi,
2006; Kokusho et al., 2011). Within three days after the mainshock, more than 900
landslides were induced by the earthquake(Chigira and Yagi, 2006;Kokusho et al.,
2014). After the earthquake sequences, a very small surface rupture was found along a
previously unmapped northern extension fault zone. The length of the surface rupture
was about 1 km (Maruyama et al., 2007). The surface slip of the Mid-Niigata
earthquake event was also very small (< 20 cm of vertical displacement). In addition,
the surface rupture zone is also far away from the epicenter zone, where the seismic
landslides have concentrated distribution (Sato et al., 2005), i.e., the study area of
seismic landslides susceptibility did not contain the surface rupture zone. So in this
study, we consider the surface rupture zone has little effect on the formation of seismic
landslides and regard the earthquake as a buried-rupture earthquake.
**3. Datasets collection**
**3.1 landslide inventory**
In this study, the assessment of seismic landslides susceptibility is performed on two
scales, the large area, and the epicenter area. As shown in Fig. 1, the large-scale area is
22 km wide (east to west), and 40 km long (north to south). The total area of the large-
scale area is about 880 km2. The epicenter area is 7 km long (north to south), and 9 km
wide (west to east). The total area is about 56 km2. The epicenter area is located in the
bordering area between Nagaoka City and Ojiya City.





Many methods have been utilized to set up landslide inventory maps, including satellite
image interpretation, aerial photography, field survey and historical landslide records
(Vařilová et al., 2015). In this research, the landslide inventory map was interpreted
from satellite image data and then checked by field survey data (Kokusho et al., 2009;
Kokusho 2008). As shown in Figure 1a, a totally of 957 landslides locations were
recorded in the large-scale area, most of which are distributed in the mountainous area
around the epicenter area and spread to the northeastern mountainous area. There are
also some landslides located in the eastern and southern mountain areas. The landslides
inventory map of the epicenter area is also shown in Fig. 1b.

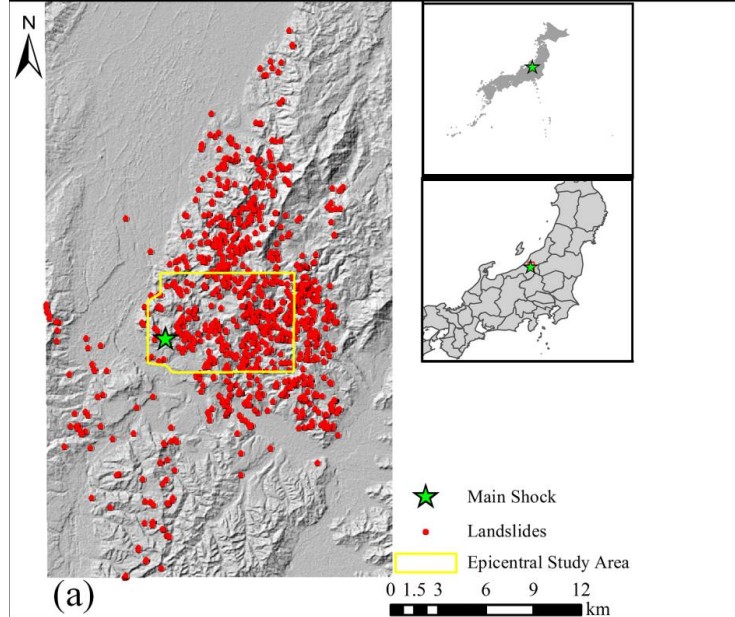



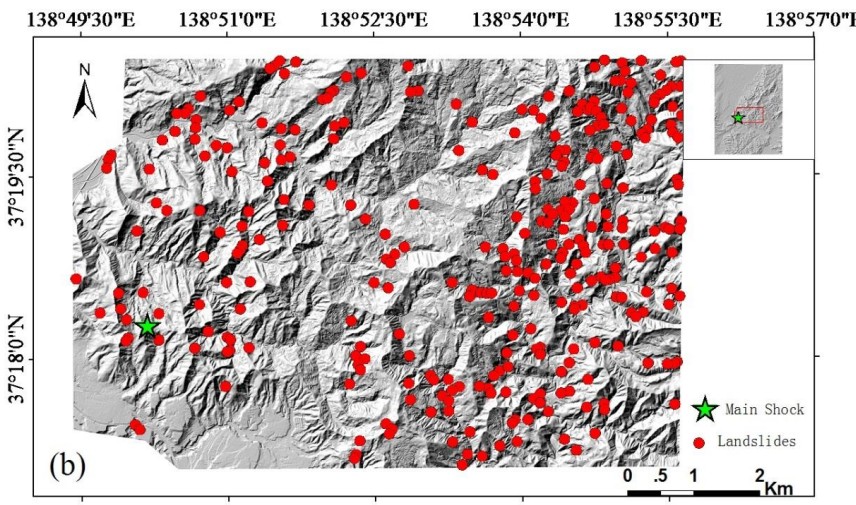


Fig.1 Locations of landslides in the study area (a) Large- scale area (b) epicenter area

## 3.2 Landslide influencing factors

The factors that affected the occurrence of earthquake-triggered landslide usually
include geology, topography, hydrology, climate, human activities, earthquake-related
parameters and etc. Based on the availability of data and impacted factors used in
previous studies (Reichenbach et al., 2018), seven landslide factors influencing
(lithology, elevation, slope, slope aspect, surface curvature, peak ground acceleration
and the distance from the road) are take into consideration for landslide susceptibility
analysis for the large-scale area. In the later analysis in the epicenter area, coseismic
ground deformation was added as an influencing factor.
Lithology directly determines the physical and mechanical properties of the slope,
which have a direct impact on slope stability. The lithology data used in this paper is
redrawn from the 1: 50000 geological map of Nagaoka and Ojiya by the Geological



Survey of Japan's Ministry of International Trade and Industry. There are ten different
lithology groups in the large-scale area (Table 1) and eleven different lithology groups
in the epicenter area (Table 2). The lithology maps in the large-area and epicenter area
are shown in Fig. 2a and Fig. 3a.
Table 1 Lithological distribution in the large-scale area.

| Category | Lithology |
|---|---|
| S | Gonglomerate with mudstone |
| G | Gonglomerate with sandstone |
| SM | Sandstone with silt |
| M | Sandstone with mudstone |
| Vs | Volcanic rock |
| Ms | Mudstone |
| Shs | Shale |
| A | Residual soil |
| Ss | Sandstone |
| Gs | Gonglomerate |

Table 2 Lithological distribution in the epicenter area area.

| Category | Lithology |
|---|---|
| QHd | Accumulation of Holocene |
| QPt | Accumulation of Pleistocene |
| QPl | Ancient landslide deposits of Pleistocene |
| QPu | Gonglomerate of Pleistocene |
| NPw | Gonglomerate of Pliocene |
| NPs | Sandy mudstone of Pliocene |
| NPu | Mudstone of Pliocene |
| NPk | Mudstone with sandstone of Pliocene |
| Nv | Volcanic rock of Pliocene |
| NMs | Shale of Miocene |


The elevation also affects the occurrence of seismic landslides (Hasegawa et al., 2009).
The elevation has bee regarded as a key factor determining gravitational potential


energy of terrain. The elevation data used in this paper is generated from the 30 m
resolution DEM data obtained from ASTER Global Digital Elevation Model (ASTER
GDEM). The elevations maps of the large-scale area and the epicenter area are shown
in Fig. 2b and Fig. 3b, respectively.
The slope angle has a direct impact on slope stability that determines the ratio of anti-
sliding force to sliding force. The slope angle in the study area ranges from 0° to 57.82°
as shown in Fig.2c for the large-area. The Fig.3c shows the distribution of slope angle
in epicenter area. The 0° slope angle means a flat area. The west part of the large-scale
area is almost flat area, whereas the mountains mainly spread from NE direction to SW
direction.
The influence of slope aspect on the stability of slope is multifaceted. Different slope
directions have different influences of solar radiation and rainfall on the slopes that
control the moisture of terrain that affects landslide occurrences. According to previous
studies (Hong et al., 2017; Pham et al., 2016; Xu et al., 2012a), the slope aspect is
divided into nine groups. The slope aspect maps of the large study area and the epicenter
area are shown in Fig. 2d and Fig. 3d, respectively and the P and FL means the flat area..
Surface curvature determines the pooling and dispersion of surface water and affects
the strength and stability of rocks and soils. In addition, there is a strong correlation
between soil thickness and surface curvature due to soil sedimentation caused by the
water flow. The surface curvature distributions in large-scale and epicenter area are
shown in Fig. 2e and Fig. 3e, respectively.
The peak ground acceleration (PGA) of the earthquakes is the maximum absolute value




of the acceleration of the surface soil in an earthquake. Since the inertia forces generated
by the earthquakes are important causes of the earthquake-triggered landslides, the PGA
is generally chosen as the impact factor of landslides susceptibility. The distribution of
peak accelerations in the large-scale area and the epicenter area is shown in Fig. 2f and
Fig. 3f, respectively.
Human activities have also greatly impacted the topography features. Road
construction not only produced a new steep cutting slope but also caused a great
disturbance to the original slope. Therefore, the distance to the road is taken into
account in the assessment of landslides susceptibility. In this study, the locations of
high-grade roads like expressway were interpreted from the satellite image. The
distances to road map were divided into seven classes (0–50, 50–100, 100–200, 200–
300, 300–400, 400-500 and >500 m). The distances to road maps of the large-scale area
and the epicenter area are shown in Fig. 2g and Fig. 3g, respectively.

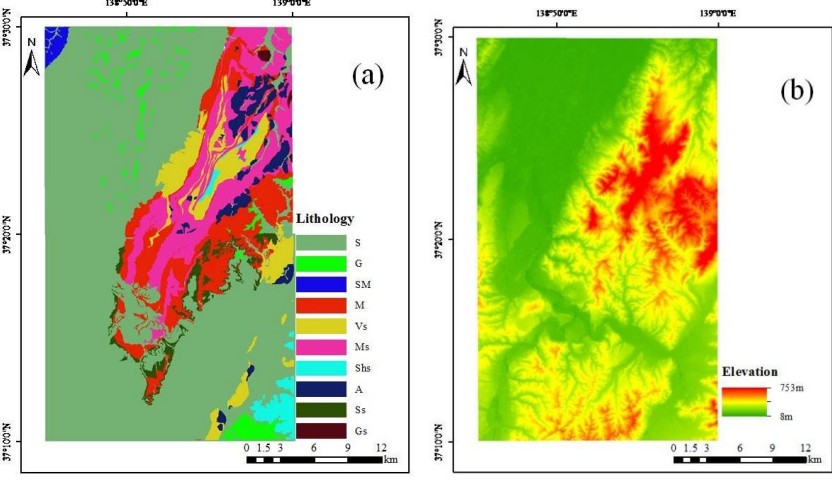




Natural Hazards
and Earth System

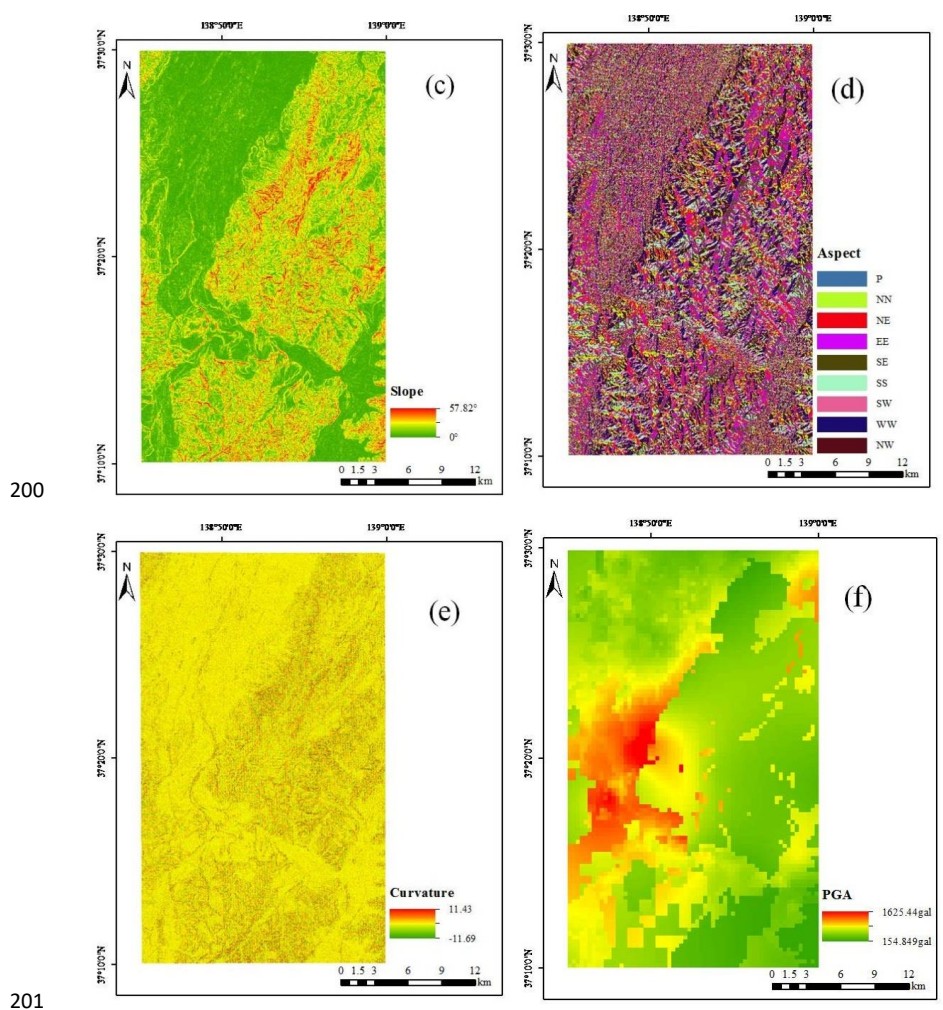





Fig.2 Landslide controlling factors of the large area, (a) lithology; (b) elevation; (c) slope

degree; (d) aspect; (e) profile curvature; (f)PGA; (g) distance to roads


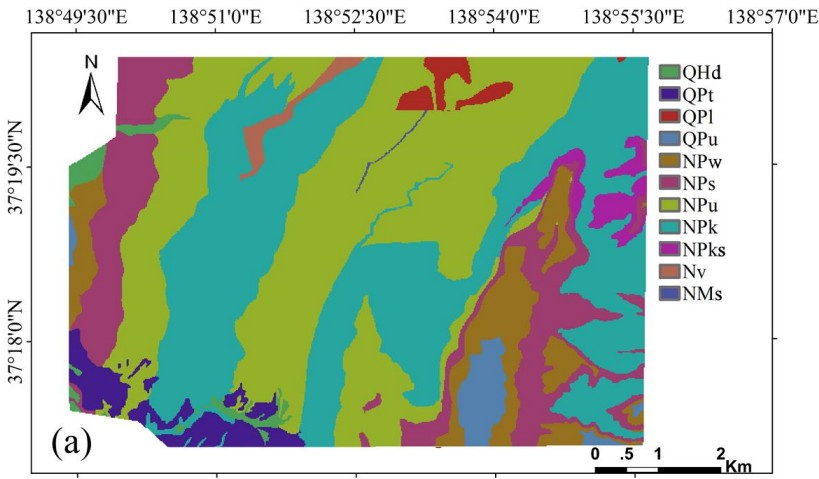



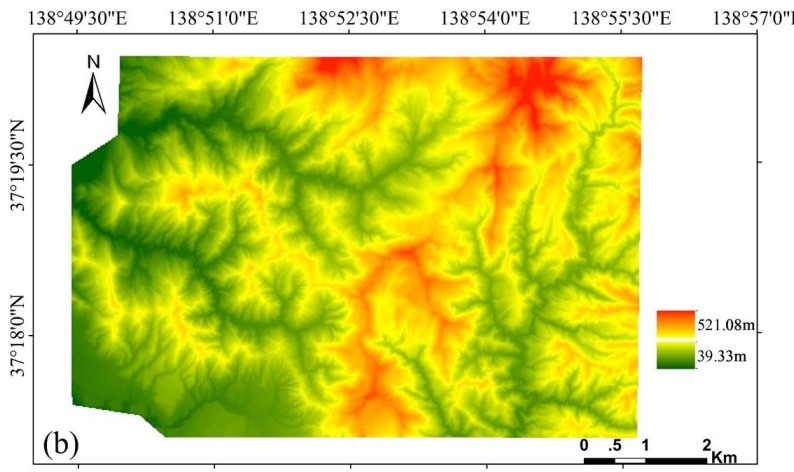


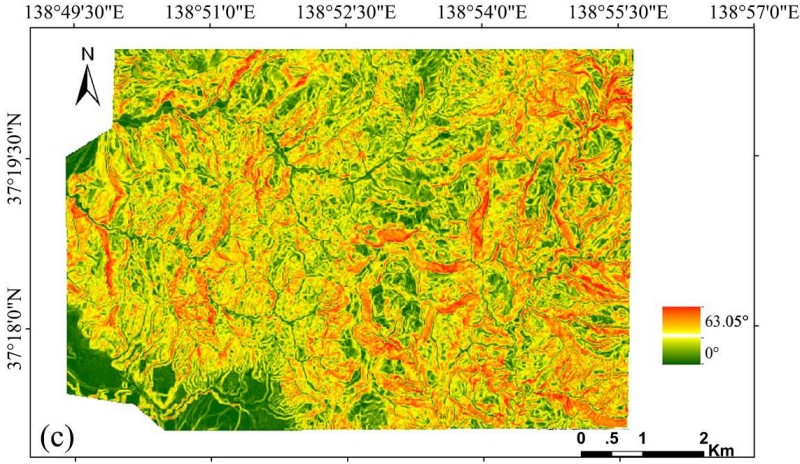




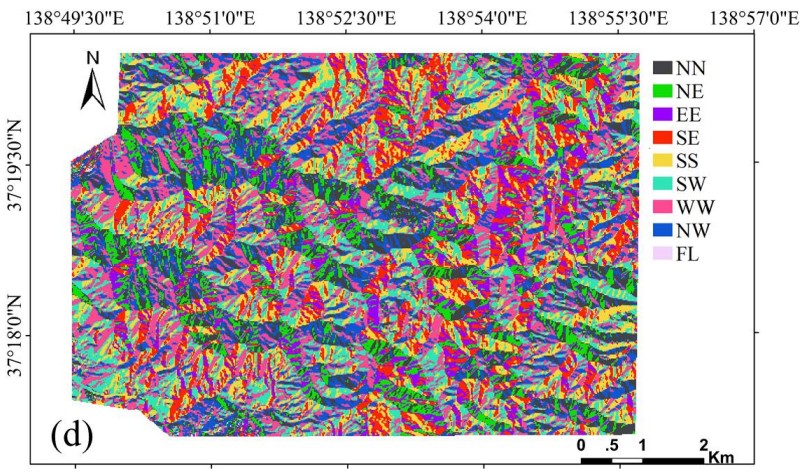


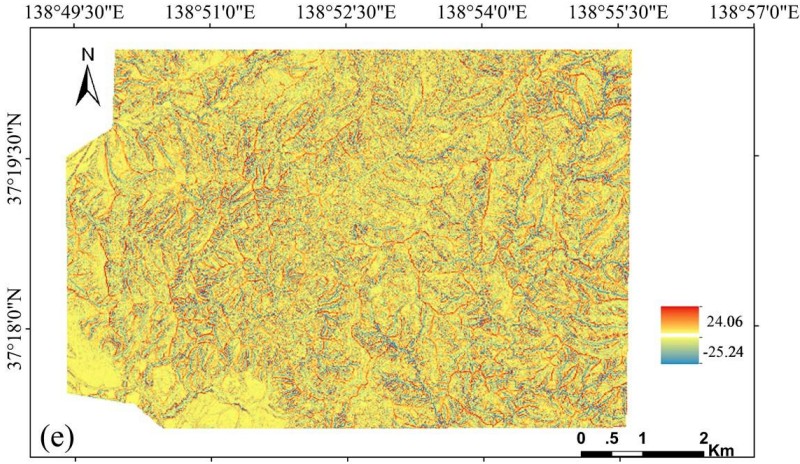


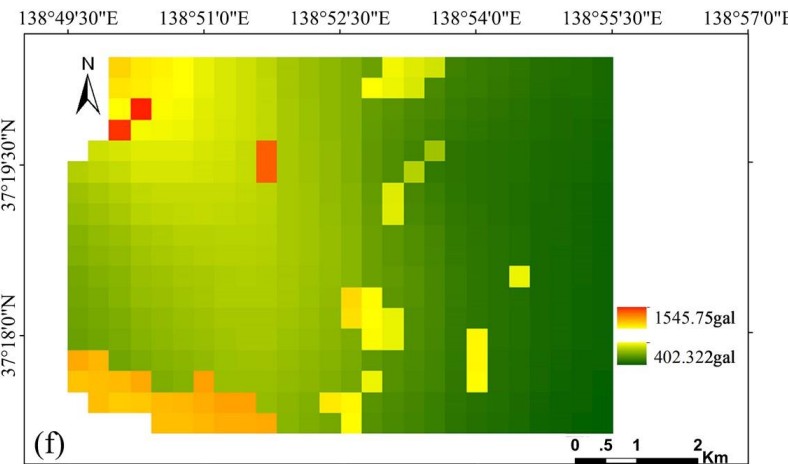


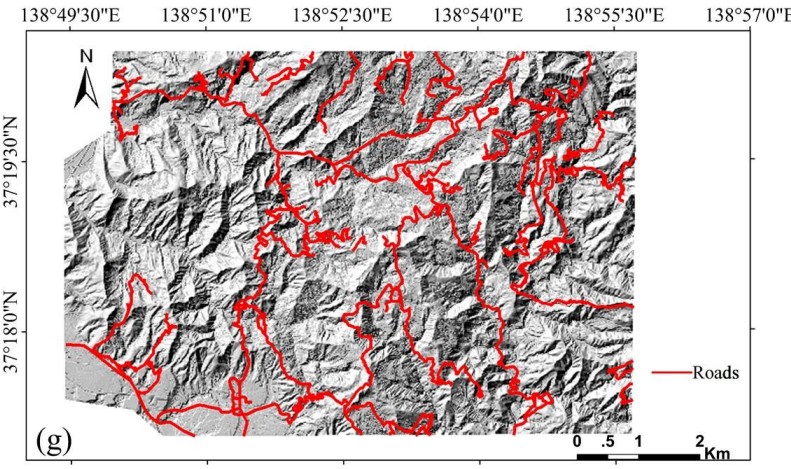


Fig. 3 Landslide controlling factors of the epicenter area, (a) lithology; (b) elevation; (c)

slope degree; (d) aspect; (e) profile curvature; (f) PGA; (g) distance to roads

For the earthquakes with surface ruptures, the previous researches show that there is a
clear connection between the landslide distribution and the distance to the rupture zone
(Xu et., al 2012b; Xu, 2014), which means the distance to the surface rupture could be
used as an influencing factor. However, for the buried-rupture earthquakes, as the very


short or no surface rupture is exposed, it is difficult to establish the relationship between
the distribution of landslides and surface rupture. Therefore, it is necessary to introduce
new influencing factors to improve the accuracy of landslides susceptibility analysis
for buried rupture earthquakes.
The coseismic ground deformation characterizes the absolute permanent ground
deformation before and after the earthquake and it has been demonstrated tha there is
a good correlation between landslides distribution and the values of coseismic ground
deformation(Chang et al., 2005; Zhao et., al 2014). Therefore, the coseismic ground
deformation could make up for the disadvantage of the losing surface rupture in the
assessment of seismic landslide susceptibility to a certain extent. The coseismic ground
deformation can be obtained by decomposed high-resolution DEM before and after the
earthquake (Zhang et al., 2010; Zhao et al., 2014). Fig. 4 illustrates the description of
landform changes in Lagrangian and Eulerian manners. Supposing that a small patch $i$
of the ground surface with one particular node mapped on it is inclined in East-West ($x$)
and North-South ($y$) directions, $\Delta z_i^e$ is expressed in terms of the Lagrangian vector $\{\Delta x_i^l$
$\Delta y_i^l \ \Delta z_i^l\}$ of the movement of the patch as:
$$\Delta z_i^e = \{t_{x,i} \quad t_{y,i} \quad 1\} \cdot \{\Delta x_i^l \quad \Delta y_i^l \quad \Delta z_i^l\}^T. \qquad (1)$$

where $t_{x,i}$ and $t_{y,i}$ are tangents of the patch plane in $x$ and $y$ directions, respectively.
Taking three adjacent patches, i1, i2 and i3 in a triangle, and using the displacement of
its center $\{\Delta x_i^l \ \Delta y_i^l \ \Delta z_i^l\}$ as the representative displacement vector of the triangle, the
following simultaneous equations are to be satisfied





$$\begin{Bmatrix} \Delta z^{e}_{i1} \\ \Delta z^{e}_{i2} \\ \Delta z^{e}_{i2} \end{Bmatrix} = \begin{bmatrix} t_{x,i1} & t_{x,i1} & 1 \\ t_{x,i1} & t_{x,i1} & 1 \\ t_{x,i1} & t_{x,i1} & 1 \end{bmatrix} \begin{Bmatrix} \Delta x^{l}_{i} \\ \Delta x^{l}_{i} \\ \Delta x^{l}_{i} \end{Bmatrix} = T \bullet \begin{Bmatrix} \Delta x^{l}_{i} \\ \Delta x^{l}_{i} \\ \Delta x^{l}_{i} \end{Bmatrix}. \quad (2)$$

An assumption that the triangle undergoes a rigid body translation is used in the
formulation above. The inclination of the moving plane (plane i1) is essential for
calculating $t_{x,i}$ and $t_{y,i}$. Suppose the equation of the moving plane is expressed as:
$z = ax + by + c$   (3)
where a= $t_{x,,i1}$=tan$\theta_{x,,i1}$, b= $t_{y,i1}$=tan$\theta_{y,i1}$

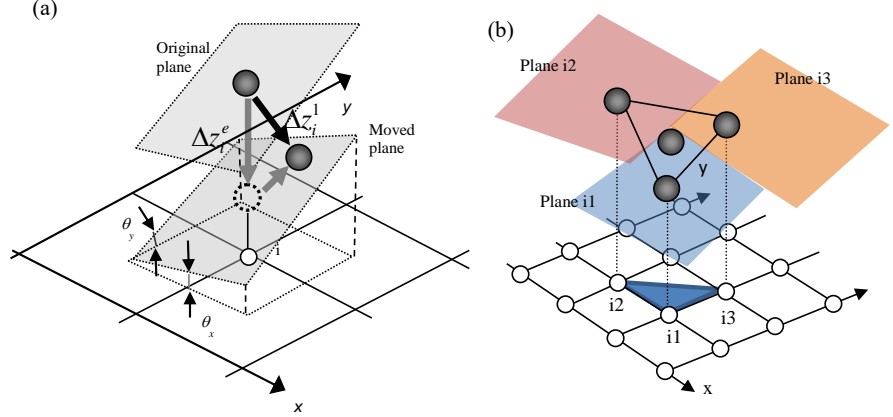


Fig.4 Description of landform change in the Lagrangian and Eurlaian manners, (a)scheme of one
point (b) scheme of three-point
Zhao et al., (2012) provides a more rigorous solution method, including the definition
of a nominal plane, the improvement of DEM comparability and matrix condition test.
In this study, we used the method that is proposed by Zhao et al., (2012) to calculate
the vacoseismic ground deformation. Note that the decomposition algorithm requires
high resolution (2m) DEM. In this study, only the epicenter area was scanned via
airborne LiDAR in 2003 and 2007, respectively. Thus, the coseismic ground
deformation is added as an influencing factor for the epicenter area only. The
distribution of coseismic ground deformation in the epicenter area is shown in Fig. 5.

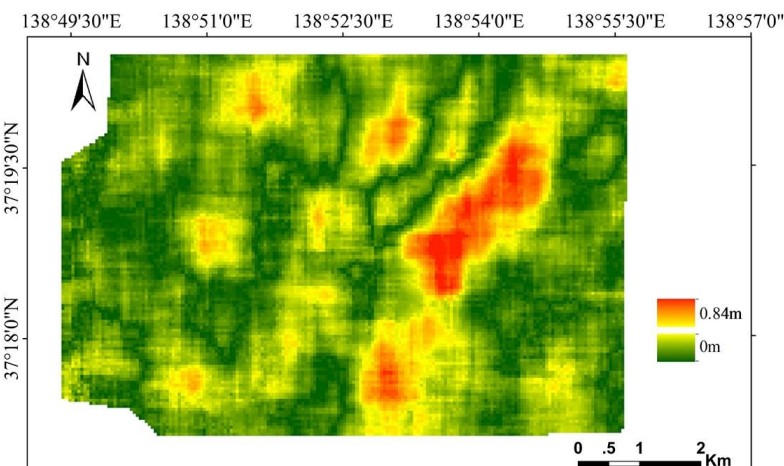


Fig. 5 The distribution of coseismic ground deformation in the epicenter area.


## 3.3 Landslides data preparation


In this study, the numbers of landslide points and non-landslide points are sampled at a
ratio of 1:1.2 for the large-scale area. A total of 1117 non-landslide points data were
randomly selected in the non-landslide area. Subsequently, 70% of the landslide points
and non-landslide points were selected randomly from the landslide inventory map as
the training dataset, with the rest as the testing dataset. In order to get optimum results,
we randomly selected the sample points (landslides points and non-landslide points) for
10 times respectively. For different selection, the training and testing samples are
different, but the numbers of sample points are the same. In the epicenter area, as the
used the method to calculate the coseismic ground deformation needs high-resolution
DEM, the whole epicenter areas were converted into 2 m pixels. The total number of




the pixels is 555324, and the number of seismic landslide pixels is 45852. Similarly,
70% of the landslide pixels and non-landslide pixels were selected randomly as the
training dataset, with the rest 30% as the testing dataset.
**4 Methodology**
**4.1 Logistic regression**
Logistic regression is suitable for describing the relationship between categorical
outcome (landslide or non-landslide) and input variables (landslide affecting factors).
The principle of the LR is to analyze the spatial relationship between the landslides
affecting factors and the occurrence of a landslide. The results of the regression usually
can be interpreted as the probability which is constrained in the interval between 0 and

1.

The LR is indicated by an equation of the form:
$$Y = f(P) = \ln(\frac{P}{1-P}) = \beta_0 + \beta_1 X_1 + \beta_2 X_2 + \ldots + \beta_n X_n \quad (4)$$
where $Y$ represents outcome variables (landslide or non-landslide), $X = X_1, X_2\ldots X_n$
represents input variables, $n$ is the $n$ $th$ landslide affecting factor, $\beta_0$ is the intercept
condition, $\beta_1, \beta_2\ldots \beta_n$ are the regression coefficients (Tu, 1996).
The SPSS 10.0 was used to conduct the LR analysis to predict the correlation between
the occurrence of landslide and landslide affecting factors. The regression coefficients
were then obtained.
The probability of a landslide event ($P$) can be determined from the following equation:




$$P = P(Y / X) = \frac{e^{\beta_0 + \beta_1 X_1 + \beta_2 X_2 + ... + \beta_n X}}{1 + e^{\beta_0 + \beta_1 X_1 + \beta_2 X_2 + ... + \beta_n X}} \quad (5)$$
The probability values change from 0 to 1, with 0 indicating a 0% probability of
landslide occurrences and 1 indicating a 100% probability.

## 4.2 Artificial neural networks (ANN)

ANN model has many advantages by comparing with other models (Yilmaz, 2009a).
ANN could process the imprecise and fuzzy data without any assumptions. The ANN
model with the most frequently used back-propagation BP algorithm (Pradhan and Lee,
2010b) is used in this paper.
The model mainly consists of one input layer, several hidden layers and an output layer.
There are usually two stages for using ANN, the training stage and classifying stage.
During the training stage, the hidden and the input layer neurons handle their inputs by
a corresponding weight, sum the product, and then deal with the sum using a nonlinear
transfer function to generate a result. During the classification period, the ANN predicts
a target value by adjusting the weights in accordance with the errors between the actual
output values and the target output ones and make the difference minimum.
In this study, the number of hidden layer nodes is calculated by Eq 6. (Yilmaz, 2009a).
$N_h = 2N_i + 1$                                   (6)
Where $N_i$ is the number of input nodes and $N_h$ is the number of hidden nodes.
Then, a three-layer network with one input layer (7 neurons), one hidden layer (15
neurons) and one output layer was used in the large-scale area. In the epicenter area, a
three-layer network consisting of one input layer (8 neurons), one hidden layer (17
neurons) and one output layer was utilized. It is important to decide the initial weight
range as influencing the convergence of the model. In this study, the initial weights
were randomly selected from a small range of [-0.25 to 0.25] as proposed by Yilmaz,
(2009b).

## 4.3 Support vector machine (SVM)

The SVM model employs nonlinear transformations of the covariates into a higher
dimensional feature space. The two main principles of SVM are the optimal
classification hyperplane and the use of a kernel function. (Yao et al., 2008).
The detailed of a two-class SVM model is described as follows. Given a set of linear
separable training vectors $x_i$ ($i$=1,2…n) that consist of two categorical outcomes
(landslide or non-landslide denoted as $y$= ±1), the purpose of the SVM is to find an $n$-
dimensional hyperplane differentiating the two categories by the maximum gap.
Mathematically, the gap $\frac{1}{2}\|w\|^2$ could be minimized subject to the following constraints
$$y_i((w \cdot x_i) + b) \geq 1 \tag{7}$$
where $\|w\|$ is the norm of the normal of the hyperplane, $b$ is a scalar base, and (·)
denotes the scalar product operation. Using the Lagrangian multiplier, the cost function
can be defined as:

$$L = \frac{1}{2}\|w\|^2 - \sum_{i=1}^{n} \lambda_i \left(y_i((w \bullet x) + b)\right) \geq 1 \tag{8}$$


where $\lambda_i$ is the Lagrangian multiplier. The solution can be obtained by the dual
minimization of Eq. (8) with respect to $w$ and $b$.
In this study, the two-class SVM method was used due to its good performance in



landslide susceptibility analysis (Yao et al. 2008; Yilmaz 2010)

## 5. Model performance validation for large-scale area

### 5.1 Training and validating the statistical models

In this study, the performances of three models (LR, ANN and SVM) for the large-scale
area were validated using receiver operating characteristic (ROC) curve. The area under
the curve (AUC) indicates how good the statistical model is. It means the model has a
perfect performance when the AUC value equals to 1. A higher AUC value indicates
better performance of the statistical model.
Because each sample datasets are selected randomly, the landslides susceptibility
calculated by the same model is not the same. In order to determine the best model, the
models are utilized for ten times analyses of randomly selected datasets, respectively.
For different analyses, the training and testing samples are different. For the same
analyses, the training samples and testing samples are the same for all three models.
The area under the ROC curve (AUC) of each analysis was compared to explore the
difference of three methods. The results are shown in Table 3.










Table 3 The AUC value of different models in large-scale area

| Number | 1 | 2 | 3 | 4 | 5 | 6 | 7 | 8 | 9 | 10 | Statistical value | |
|---|---|---|---|---|---|---|---|---|---|---|---|---|
| Model | % | % | % | % | % | % | % | % | % | % | Average value | Variance value |
| LR | 82.0 | 81.6 | 82.3 | 80.2 | 81.4 | 80.2 | 81.4 | 80.7 | 81.6 | 82.0 | 81.3 | 0.49 |
| ANN | 83.3 | 82.4 | 83.6 | 81.3 | 82.1 | 82.0 | 82.1 | 82.3 | 82.8 | 83.1 | 82.5 | 0.44 |
| SVM | 80.8 | 80.9 | 81.8 | 80.1 | 80.7 | 79.4 | 80.4 | 80.5 | 80.5 | 81.8 | 80.7 | 0.47 |


Table 3 shown as the ANN model performed the best among the three models with the
highest AUC value, and the accuracy of the SVM model was worst. Based on the
maximum AUC values of ten simulations, the ANN simulation result was also the best
(83.6%). The average value and variance of the ANN model were 82.5% and 0.44%,
which was better than the LR and SVM models. It means the robustness of the ANN
model is better than LR and SVM models.
Yilmaz (2009a) used three models including frequency ratio (FR), ANN and LR to
generate the landslide susceptibility maps of Kat County (Tokat–Turkey). The result
showed the ANN model performed better than other models. In Yilmaz (2010), four
different models including conditional probability (CP), LR, ANN, and SVM models
were utilized to assess the landslide susceptibility of Koyulhisar (Sivas, Turkey). The
results also showed the performance of the ANN model was best. Some other research
also showed the ANN model performed more accurate than other models (Yesilnacar
and Topal 2005; Pradhan and Lee 2010c; Gómez and Kavalgu 2005). We consider the



ANN model performed better than the other models because it has a good global
searching ability and can learn the near-optimum solution without the gradient
information of error functions. As there about a total of 2000 samples in the calibration
and validation set. Large numbers of samples in the calibration stage will lead to
sufficient training of the model and establish an appropriate structure of ANN model.
So the ANN model performs well on the condition those large numbers of samples were
available. For any algorithm, the quantity and quality of samples have key impacts on
the accuracy of the predicted results the algorithm makes.

## 5.2 Development of landslide susceptibility maps

In this study, all three models have been used to calculate the landslide susceptibility
index (LSI) of each point, then generating the landslide susceptibility maps. There are
several mathematical methods including quantiles, natural breaks, standard deviation,
equal intervals, and descending area percentage to be reclassified the LSI (Ayalew et
al., 2004). Among the above methods, the descending area percentage technique is the
most widely used. In this study, the descending area percentage technique was used.
The landslide susceptibility maps were constructed into four classes: low (40%),
moderate (30%), high (20%), and very high (10%). The landslide density was used to
assess the performance of landslide susceptibility maps. The landslide density (LD) is
defined as the ratio of the numbers of landslide and the area of each susceptible class.
The calculated landslide densities by using the three different models are shown in
Table 4. It can be observed that all maps present good spatial predictions of landslides





as landslide density is ascending from very low to very high class (Yilmaz, 2009b). The
results using the ANN model show that the very high class contains 42.01% of the total
landslides, however, it only covers 9.95% of the total study area and the LD of the very
high class was 4.59. In comparison, the low classes only contain 3.34% landslides,
however, it covers 40.15% area and the LD of the low class was 0.09. This indicates
that the ANN model performed well in susceptibility classification as it fits well with
the landslide inventories.
Table 4 The distribution of different classes area obtained by different methods

| Model | Class | Area (km$^2$) | Times of landslides occurrence | Percentage of each susceptible class area (%) | Percentage of landslides in each susceptible class (%) | Landslides density (times/km$^2$) |
|---|---|---|---|---|---|---|
| LR | Very high | 87.61 | 387 | 9.95 | 40.44 | 4.42 |
| | high | 175.55 | 350 | 19.95 | 36.57 | 1.99 |
| | moderate | 263.60 | 171 | 29.95 | 17.87 | 0.65 |
| | low | 353.35 | 49 | 40.15 | 5.12 | 0.14 |
| SVM | Very high | 87.71 | 473 | 9.97 | 49.43 | 5.39 |
| | high | 175.95 | 286 | 19.99 | 29.89 | 1.63 |
| | moderate | 264.16 | 126 | 30.01 | 13.17 | 0.48 |
| | low | 352.28 | 72 | 40.03 | 7.52 | 0.20 |
| ANN | Very high | 87.60 | 402 | 9.95 | 42.01 | 4.59 |
| | high | 175.54 | 352 | 19.95 | 36.78 | 2.01 |
| | moderate | 263.60 | 171 | 29.95 | 17.87 | 0.65 |
| | low | 353.37 | 32 | 40.15 | 3.34 | 0.09 |


The landslides susceptibility maps of different methods are shown as Fig. 6. The
analyses result of LR, SVM and ANN models are very close. The epicenter area is a
very high susceptible area, the northeast and the southwest mountain area are high and
very high susceptible areas respectively, and the northern plains area is basically
distributed with low susceptible class. The susceptibility map of the ANN model shows
the high susceptible areas and low susceptible areas are more concentrated into blocks,
and zonation produced by the SVM and LR are more dispersed. Overall, all three
models could generate reasonable landslides susceptibility maps.

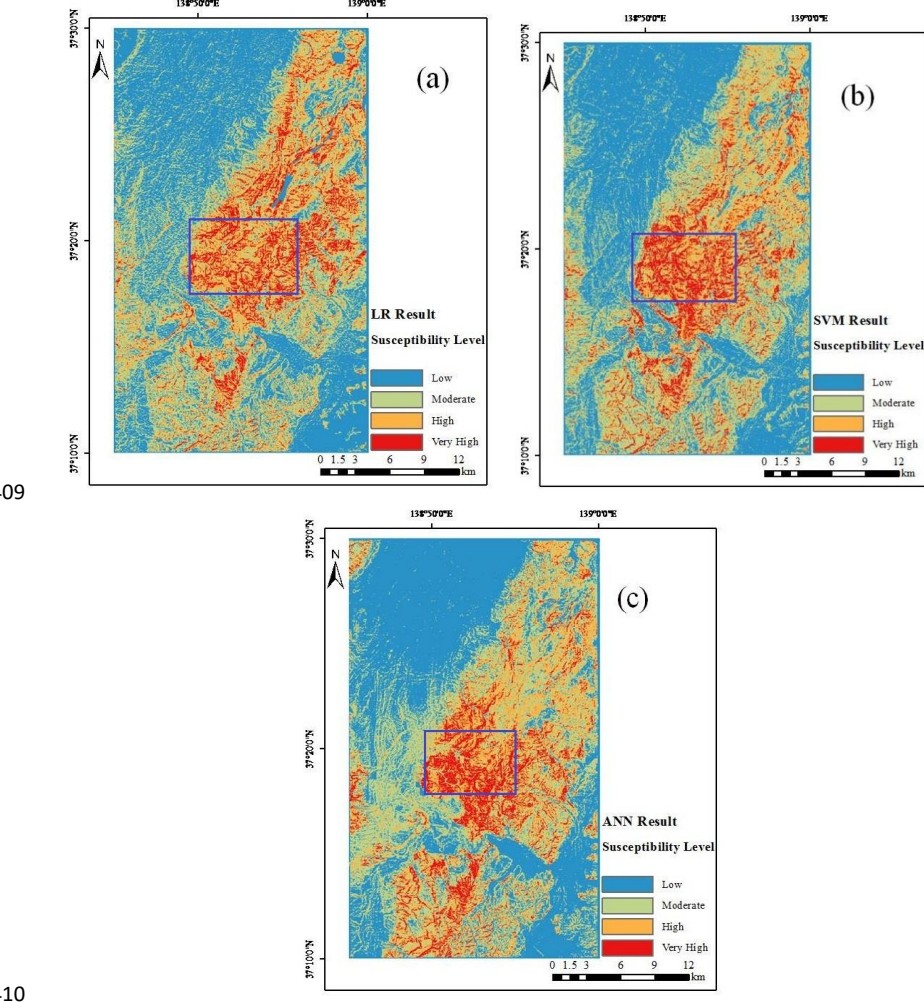

Fig. 6 Landslide susceptibility maps using different models for the large area. (a) LR model (b)
SVM model (c) ANN model



## 6. Model performance validation for epicenter area


From the landslide susceptibility map of the large-scale area, it is known that the
susceptibility level in the epicenter area is generally high. Since it is still too costly to
remediate all slopes in the approximately 60 km$^2$ area, it is necessary to further evaluate
the landslides susceptibility of the epicenter area. It can be seen in Section 5 that the
ANN model is the most suitable model for landslides susceptibility assessment in this
area. Therefore, we only use the ANN model to analyze and evaluate the landslides
susceptibility in the epicenter area.
Firstly, in order to evaluate the significance of landslides susceptibility analysis with
considering different scales. The values of AUC for the epicenter area are calculated in
two different conditions. Firstly, we calculate the values of AUC based on the
corresponding calculated LSI of the epicenter area from the large-scale (whole affected
area) datasets. Then, the values of AUC are calculated based on the calculated LSI from
the epicenter area datasets. The values of AUC of the two different conditions are shown
as Fig. 7. The results show the AUC is 56.2% based on the calculated LSI from the
large-scale datasets, on the contrast the AUC is 72.3% based on the calculated LSI from
the epicenter area solely. The results show it is necessary to assess the landslides
susceptibility under different scales.


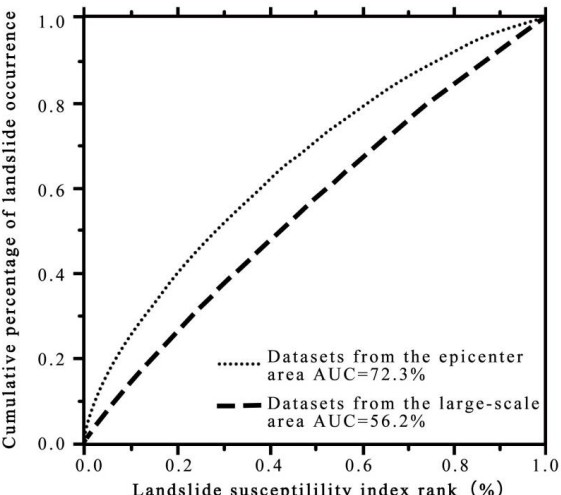


Fig.7 Analysis of the ROC curve under different scales

Then, in order to evaluate the effects of the new factor coseismic ground deformation
on the assessment of landslides susceptibility, two different situations are considered.
One situation regards the coseismic deformation as an influencing factor, whereas the
other does not. Fig.8 shows the values of AUC with considering coseismic surface
deformation or not. From Fig. 8, it could be known that the AUC is 72.3% without
considering the coseismic surface deformation, on the contrast the AUC is 76.5% with
considering the coseismic surface deformation. It means the coseismic surface
deformation has a positive effect on the assessment of landslides susceptibility.

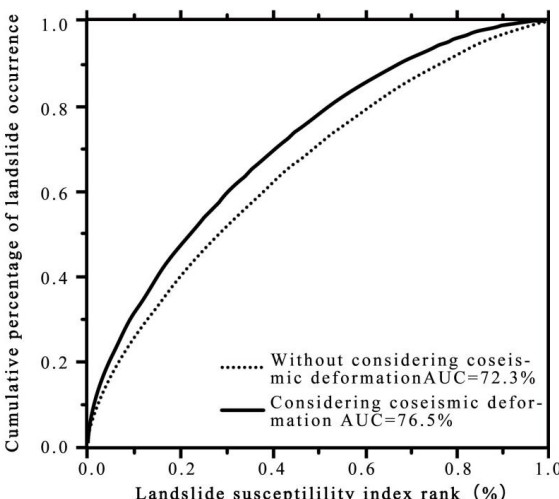


Fig.8 Analysis of the ROC curve with considering coseismic surface deformation or not


Influencing factors including lithology, elevation, slope, slope aspect, surface curvature,
peak ground acceleration, the distance from the road and coseismic ground deformation
were considered in the present study. Since the contribution of these factors to landslide
models might be different, it is necessary to quantify the effects of influential factors
on the assessment of landslides susceptibility. The Analysis of Variance method
(ANOVA) has been utilized to evaluate the predictive capability of these factors. The
factors with higher variance values indicate a higher contribution to landslide models
and vice versa. The predictive capability of eight landslide affecting factors was shown
in Table 5.







Table 5 the predictive importance of different influencing factors

| Number | Influencing factor | Predictive importance |
|---|---|---|
| 1 | Lithology | 0.213 |
| 2 | Slope | 0.207 |
| 3 | PGA | 0.169 |
| 4 | Curvature | 0.125 |
| 5 | Coseismic ground deformation | 0.093 |
| 6 | Elevation | 0.086 |
| 7 | Slope aspect | 0.057 |
| 8 | Distance to roads | 0.048 |


As Table 5 shown, lithology has the greatest impact on the occurrence of earthquake
landslides and the impact of other factors is in order of slope, peak earthquake
acceleration, curvature, coseismic ground deformation, elevation, aspect and distance
from the road. The importance of coseismic surface deformation is higher than the
elevation, aspect and distance from the road that are commonly chosen as influencing
factors in the assessment of landslides susceptibility (Reichenbach et al., 2018).
Although the earthquakes do not produce obvious ground rupture, the area with large
coseismic surface deformation indicates that the movement of the rock mass may be
further developed and the integrity of rock mass is reduced, which renders slopes prone
to landslip in future earthquakes again. Therefore, especially in the case of buried fault
earthquakes, coseismic surface deformation can be considered as an important
influencing factor in the assessment of earthquake landslides susceptibility.
Subsequently, the landslides density and landslides susceptibility map of the epicenter
area were obtained as shown in Table 6 and Fig 9. The results show that the very high
class contains 40.44% of the total landslides, however, it only covers 8.6% of the



epicenter area and the LD of the very high class was 26.54. In comparison, the low
classes contain only 5.12% landslides, however, it covers 40.15% area and the LD of
the low class was only 0.73. The landslide density is increasing gradually between low
class and very high class. This indicates that the landslide susceptibility map fits well
with the landslide inventories.
Table 6 The distribution of different classes area in the epicenter area

|  | Class | Area (km$^2$) | Landslides occurrence | Percentage of each susceptible class area (%) | Percentage of landslides in each susceptible class (%) | Landslides density (times/km$^2$) |
|---|---|---|---|---|---|---|
| ANN | Very high | 4.7853 | 127 | 8.6 | 40.44 | 26.54 |
|  | high | 10.8605 | 115 | 19.51 | 36.57 | 10.57 |
|  | moderate | 18.089 | 56 | 32.5 | 17.87 | 3.10 |
|  | low | 21.9236 | 16 | 39.39 | 5.12 | 0.73 |


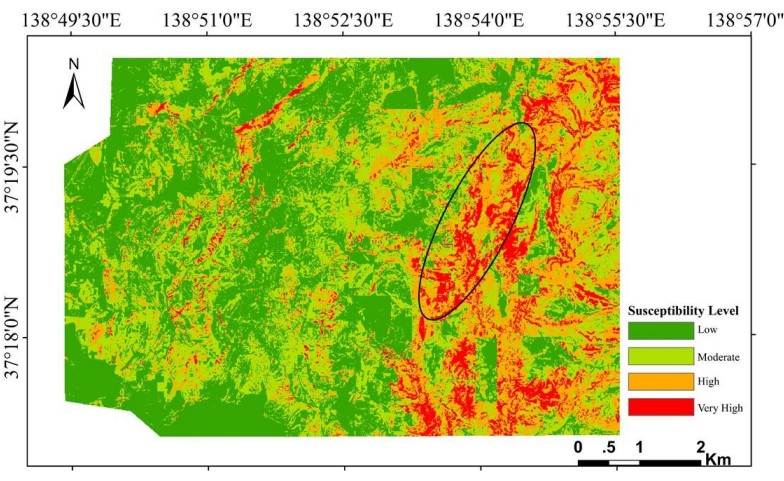


Fig. 9 Landslides susceptibility map of epicenter area
As shown in Figure 9, the very high-class area is mainly distributed along the long axis
of the ellipse in the east of the study area, and a large amount of deep-seated landslide
occurred in this area. The high susceptibility area is also distributed in the northwestern





area. The occurrence possibility of landslides in the central area and southwest plain
area is relatively low. Compared with the epicenter area parts in the landslides
susceptibility map of large-scale, the landslides susceptibility maps obtained by the
epicenter area research have a better discrimination degree, which can meet the key
prevention and control requirements in the small area.

## 491  7. Conclusion

In this paper, the LR, ANN, and SVM models are applied to generate landslide
susceptibility maps based on the 2004 Mid-Niigata earthquake-triggered landslide
inventories. Seven impact factors, such as lithology, elevation, slope, aspect, surface
curvature, peak acceleration and the distance from the road are selected as the
influenced factors. The ROC curve evaluation results clearly demonstrate that the map
obtained from the ANN model performed the best among the three models. The
variance of AUC for randomly selected datasets by ANN is also the smallest, which
means the ANN model has excellent robustness.
Therefore, the ANN model can be used for the assessment and the development of
landslide susceptibility map. Then, the significance of landslides susceptibility analysis
with considering different scales is also evaluated. The results show the AUC is 56.2%
based on the datasets from the large-scale, on the contrast the AUC is 72.3% based on
the datasets from the epicenter area solely. The results show it is necessary to assess the
landslides susceptibility under different scales. At the same time, we included the
coseismic ground deformation as the influencing factor for landslides susceptibility in



the epicenter area. The AUC increased from 0.723 to 0.765 after considering the newly
added factor. Therefore, for the buried rupture earthquake, the coseismic surface
deformation can be considered as an important factor to evaluate the susceptibility of
landslides.

## Acknowledgment

This research was supported financially by the Fundamental Research Funds for the
Central Universities (2019FZJD002). We appreciate Prof. Fuchu Dai in Beijing
University of Technology for his advice on improving the study.

## Author contributions

ZY, ZH and ZW conceived this research. ZY, ZJ and KK designed the methodology and
performed the experiments. ZY and ZW analysed the results and wrote the paper. All
authors contributed to the preparation of this paper.

## Competing interests.

The authors declare that they have no conflict of interest.

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
