# Peer review of "The assessment of earthquake-triggered landslides"

_Natural Hazards and Earth System Sciences, 2020_

## Referee Comment (RC1) · Anonymous Referee #1 · 17 Apr 2020

In the manuscript "The assessment of earthquake-triggered landslides suscepti-bility with considering coseismic ground deformation", the authors try to improve earthquake-triggered landslide susceptibility maps by introducing a new parameter that they call "coseismic ground deformation" Assuming as a study case the October 23, 2004, Mw 6.8 Niigata earthquake, the authors provide several landslide susceptibil-ity maps at two different scales, using three different statistical methods, namely, the logical regression (LR), the Support Vector Machine (SVM), and the Artificial Neural Network (ANN). The authors conclude their study, saying that the "coseismic ground deformation" parameter is an "important" factor to evaluate the susceptibility of land-slides. In my opinion, the small increase of the area under the receiver operating

characteristic (ROC) curve, i.e., from approximately 0.72 to 0.77, obtained by introducing the "coseismic ground deformation" parameter, does not support the conclusion stated by the authors. Such a small improvement is primarily affected by the generally low resolution of the other parameters introduced in the analysis. In particular, one of the most critical parameters, i.e., the lithological map, present a spatial resolution that could be not acceptable for a small scale analysis of the epicentral area. Another critical factor, i.e., the peak ground acceleration, is strongly scattered over the study area, and it presents an unacceptable resolution for the small scale study of the epicentral area. Even the "coseismic ground deformation" parameter and its definition are not explicit. The authors provide a map (Figure 5) given by the difference of two Lidar surveys performed in 2003 and 2007. The authors do not specify the orientation of the computed ground deformation (subsidence? Uplift?....) nor describe the "expected" coseismic ground deformation concerning the faulting mechanism. Moreover, the map shown in Figure 5 covers approximately four years; thus, it could be affected by ground movements that are not related to the earthquake. It is widely acknowledged that the most critical factors that affect earthquake-triggered landslides are the lithology, the slope, and the PGA. Regarding the latter, slope stability is affected not only by the PGA of the mainshock but also by the PGA of the several aftershocks, which always follow the main event. This aspect is neither introduced not discussed in the analysis. Therefore, according to the several uncertainties related to the selection, calibration, and description of the parameters affecting landslide susceptibility, the small increase in AOC associated with the introduction of the "coseismic ground deformation" parameter is not significant. According to the comments above, the manuscript is not acceptable for publication.
* * *

---

## Referee Comment (RC2) · Anonymous Referee #2 · 27 Apr 2020

In this manuscript, authors present the results of statistical analyses done to the distribution of landslides induced by the Mid-Niigata earthquake (2004), Mw 6.8. Three different statistical methods (logistic regression, Artificial Neural Network and Support Vector Machine) are applied to landslide inventory at two different scales: regional and near field. In this last case, coseismic ground deformation is considered as an influencing factor in the susceptibility analysis. From the analyses, the ANN method gives the best results.

The objective of the paper is to analyze the importance of the coseismic ground deformation to explain landslide distribution and the benefits of using it when preparing

susceptibility maps.

The paper is properly organized and most of figures and tables are of interest.

Regarding the main objective of the paper, I miss a reflection by the authors about the true usefulness of the parameter in question in the preparation of susceptibility maps. As the authors point out in the Introduction, these maps constitute the main tool that our society has to establish the areas prone to suffer seismic-induced landslides, and thus define an appropriate use (or restrict their occupation) of the territory. However, the parameter that constitutes the center of the article, the coseismic ground deformation, is a parameter that can only be evaluated afterwards, that is, once the earthquake has occurred. So what real use does it have? Personally, I see this parameter, as well as the distance to the surface of rupture, useful for subsequent studies, to explain why instabilities have occurred in certain contexts or areas, but not to predict their occurrence. In fact, the difference in AUC when considering/not considering this parameter is less than 5%.

---

## Author Comment (AC1) · 17 May 2020

**Response to Review Comments**

Title: **The assessment of earthquake-triggered landslides susceptibility with considering coseismic ground deformation**

First of all, the authors are grateful to the reviewer, who offered many constructive suggestions to enhance the manuscript. With this reply we hope to provide adequate answers to the comments of the reviewers. This is done in a point-by-point fashion below.

**Responses to the Comments Raised by Reviewer #1**

1. In the manuscript "The assessment of earthquake-triggered landslides susceptibility with considering coseismic ground deformation", the authors try to improve earthquake-triggered landslide susceptibility maps by introducing a new parameter that they call "coseismic ground deformation" Assuming as a study case the October 23, 2004, Mw 6.8 Niigata earthquake, the authors provide several landslide susceptibility maps at two different scales, using three different statistical methods, namely, the logical regression (LR), the Support Vector Machine (SVM), and the Artificial Neural Network (ANN). The authors conclude their study, saying that the "coseismic ground deformation" parameter is an "important" factor to evaluate the susceptibility of landslides. In my opinion, the small increase of the area under the receiver operating characteristic (ROC) curve, i.e., from approximately 0.72 to 0.77, obtained by introducing the "coseismic ground deformation" parameter, does not support the conclusion stated by the authors. Such a small improvement is primarily affected by the generally low resolution of the other parameters introduced in the analysis.

**Authors' reply:**

Many thanks for your comments. In this study, we aim to show the importance of

"coseismic ground deformation" in landslides mapping by comparing the values of AUC in the condition of considering and without considering coseismic ground deformation. The only difference between the two conditions was adding the coseismic ground deformation. It means the resolution of the other parameters in the two condition are exactly the same. So, the authors considered that the effects of other parameters resolution on the calculated results should be very small.

In addition, in order to evaluate the effects of the coseismic ground deformation on the assessment of landslides susceptibility, the Analysis of Variance method (ANOVA) has been utilized to evaluate the predictive capability of used conditional factors. The factors with higher variance values indicate a higher contribution to landslide models and vice versa. The predictive capability of eight landslide affecting factors was shown in Table 1.

Table 1. the predictive importance of different influencing factors

| Number | Influencing factor | Predictive importance |
|--------|--------------------|-----------------------|
| 1 | Lithology | 0.213 |
| 2 | Slope | 0.207 |
| 3 | PGA | 0.169 |
| 4 | Curvature | 0.125 |
| 5 | Coseismic ground deformation | 0.093 |
| 6 | Elevation | 0.086 |
| 7 | Slope aspect | 0.057 |
| 8 | Distance to roads | 0.048 |

From Table 1, it could be found the coseismic ground deformation ranked the fifth among eight factors. The importance of coseismic surface deformation is higher than the elevation, aspect and distance from the road. Reichenbach et al., (2018) critically review the statistically-based landslide susceptibility assessment literature by systematically searching for and then compiling an extensive database of 565 peer-review articles from 1983 to 2016. The results showed that elevation, aspect and distance from the road are commonly chosen as influencing factors in the assessment of landslides susceptibility. It means the coseismic ground deformation should be

regarded as an important factor in the assessment of landslides susceptibility.

The AUC is a commonly used indices to evaluate the model prediction performance. At present, there are no unanimous standards to assess the increment of AUC. This means it is still debated that how much increment of AUC will be regarded as significant improvement. Most studies just considered the larger value of AUC means the better performances of the model. For example, Pham et al., (2016) conducted a comparative study of five different machine learning methods for landslide susceptibility assessment. The increment of AUC value for different models was about 0.045 (0.910-0.955). Yilmaz (2010) made a comparison of landslide susceptibility mapping methods. The increment AUC value for different models was 0.019 (0.827-0.846). Pham et al., (2017a) made a comparative study of sequential minimal optimization-based support vector machines, vote feature intervals, and logistic regression in landslide susceptibility assessment. The increment of AUC value for different models was 0.044 (0.812-0.856). Pham et al., (2017b) used the hybrid integration of multilayer perceptron neural networks and machine learning ensembles for landslide susceptibility assessment. The increment of AUC value for different models was 0.01 (0.876-0.886). Aghdam et al., 2017 conducted the landslide susceptibility assessment using a novel hybrid model of statistical bivariate methods (FR and WOE) and adaptive neuro-fuzzy inference system (ANFIS). The increment of AUC value for different models was 0.03 (0.82-0.85). Tsangaratos and Ilia (2016) conducted the landslide susceptibility mapping using the certainty factor method, the Iterative Dichotomizer version 3 algorithm, the J48 algorithm and the modified Iterative Dichotomizer version 3 model. The validation results showed that the AUC values for these models varied from 0.7766 to 0.8035. Xu et al., (2012) made a comparison of different models for susceptibility mapping of earthquake triggered landslides related with the 2008 Wenchuan earthquake in China. The results showed that the AUC values for the models varied from 0.7253 to 0.801. So, comparing the increment of AUC values in this study with above mentioned studies, it may be concluded that the increasing of AUC is relatively significant. Furthermore, the AUC of the analysis in this study is relatively low, which badly requires adding more contribution factors to improve the performance.

In addition, the coseismic ground deformation will help to reveal the hidden subsurface damage. It should be noted that not all deformation will direct lead the landslides. However, the area with large coseismic surface deformation often indicates that the movement of the rock mass may be further developed and the integrity of rock mass is reduced, which renders slopes prone to landslip in future earthquakes again. Zhao et al., (2012) explored the localized coseismic deformation in Kizawa (a small village), Japan after the earthquake. The results showed the calculated coseismic deformation in Kizawa is relatively large but the landslides are sparse. However, after a detail investigation, it found that the underground structures such as tunnels and wells were severely damaged. The road alignment of the Kizawa tunnel, which was buried 30 m beneath the ground surface, was shifted sideways 1-1.5 m to east-to-southeast direction. Furthermore, two irrigation well were dislocated at 30 m and 20 m, beneath the ground, respectively. Therefore, it is highly possible that the ground underwent some subsurface damage at locations where the large coseismic deformation. Although the deformation did not form the landslides at these locations in the 2004 Mid-Niigata earthquake, as there were accumulated deformation within the rock and soil, the landslide will easily occur in the next earthquake event. According to the comments above, the coseismic ground deformation should be regarded as a useful influencing factor in the assessment of landslides susceptibility.

*Reference*
1. *Pham, B.T., Pradhan, B., Bui, D.T., Prakash, I., Dholakia, M.B. 2016. A comparative study of different machine learning methods for landslide susceptibility assessment: A case study of Uttarakhand area (India). Environmental Modelling & Software. 84, 240-250*
2. *Yilmaz, I., 2010. Comparison of landslide susceptibility mapping methodologies for Koyulhisar, Turkey: conditional probability, logistic regression, artificial neural networks, and support vector machine. Environmental Earth Sciences, 61(4): 821-836.*
3. *Pham, B.T., Bui, D.T., Prakash, I., Long, H.N. and Dholakia, M.B., 2017a. A comparative study of sequential minimal optimization-based support vector machines, vote feature intervals, and logistic regression in landslide susceptibility assessment using GIS. Environmental Earth Sciences, 76(10): 371.*
4. *Pham, B.T., Bui, D.T., Prakash, I., Prakash, I. and Dholakia, M.B., 2017b. Hybrid integration*

*of Multilayer Perceptron Neural Networks and machine learning ensembles for landslide susceptibility assessment at Himalayan area (India) using GIS. Catena 149, 52–63*

5. *Aghdam, I.N., Pradhan, B., Panahi, M., 2017. Landslide susceptibility assessment using a novel hybrid model of statistical bivariate methods (FR and WOE) and adaptive neuro-fuzzy inference system (ANFIS) at southern Zagros Mountains in Iran. Environmental Earth Sciences, 76: 237.*

6. *Tsangaratos, P., Ilia, L., 2016. Landslide susceptibility mapping using a modified decision tree classifier in the Xanthi Perfection, Greece. Landslides, 13:305–320.*

7. *Xu, C., Xu, X., Dai, F. and Saraf, A.K., 2012. Comparison of different models for susceptibility mapping of earthquake triggered landslides related with the 2008 Wenchuan earthquake in China. Computers & Geosciences, 46(3): 317-329.*

8. *Zhao, Y., Konagai, K. and Fujita. T., 2012. Multi-scale Decomposition of Co-seismic Deformation from High Resolution DEMs: a Case Study of the 2004 Mid-Niigata Earthquake. Acta Geologica Sinica(English Edition), 86(4): 1013-1021.*

9. *Reichenbach, P., Rossi, M., Malamud, D.B., Mihir, M. and Guzzetti, F., 2018. A review of statistically-based landslide susceptibility models. Earth-Scienc Reviews, 180: 60-91.*

2. In particular, one of the most critical parameters, i.e., the lithological map, present a spatial resolution that could be not acceptable for a small scale analysis of the epicentral area.

**Authors' reply:**

Many thanks for your comments. The used lithological map is the highest resolution map that the author could get. Even so, in the used lithological map, there are totally eleven different types of lithology. Bandara and Ohtsuka (2017) explored the spatial distribution of landslides induced by 2004 Mid-Niigata prefecture earthquake and only seven different lithology types were distinguished. In addition, comparing with other similar studies (Yi et al., 2019; Xu et al., 2012; Yang et al., 2014), the common number of lithology types in earthquake induced landslide susceptibility mapping were ten to fifteen. So, the authors considered the spatial resolution of the lithological map was acceptable to generate the landslides susceptibility mapping.

*Reference*
*1.Yi, Y.N., Zhang, Z.J., Zhang, W.C., Xu, Q., Deng, C. and Li, Q.L., 2019. GIS-based earthquake-triggered-landslide susceptibility mapping with an integrated weighted index model in Jiuzhaigou region of Sichuan Province, China. Natural Hazards and Earth System Science, 19, 1973–1988.*

*2. Xu, C., Xu, X., Dai, F. and Saraf, A.K., 2012b. Comparison of different models for susceptibility mapping of earthquake triggered landslides related with the 2008 Wenchuan earthquake in China. Computers & Geosciences, 46(3): 317-329.*

*3. Yang, Z.H., Lan, H.X., Gao, X., Li, L.P., Meng, Y.S. and Wu, Y.M., 2015. Urgent landslide susceptibility assessment in the 2013 Lushan earthquake-impacted area, Sichuan Province, China. Natural Hazards, 75(3): 2467-2487.*

*4. Bandara, S. and Ohtsuka, S., 2017. Spatial distribution of landslides induced by the 2004 Mid-Niigata prefecture earthquake, Japan. Landslides, 14:1877-1886.*

3. Another critical factor, i.e., the peak ground acceleration, is strongly scattered over the study area, and it presents an unacceptable resolution for the small scale study of the epicentral area.

**Authors' reply:**

Many thanks for your comments. The authors fully agree that the spatial resolution of PGA map is low. The PGA map of epicenter area is the result of back-analysis for the seismic station data. So, the resolution is relatively low. However, the used PGA map is also the highest resolution map that the authors could get and most back-analyses can offer. The authors inferred that the low resolution of PGA map is also a main reason to lead relatively lower values of AUC (0.72), as the values of AUC in large-scale is 0.82 for ANN model. However, in the other words, the low values of AUC also demonstrated the urge demand of introducing new factors to improve the assessment of landslides susceptibility. As the low resolution of PGA map is a main limitation, the authors will also discuss this in the revised manuscript.

4. Even the "coseismic ground deformation" parameter and its definition are not explicit. The authors provide a map (Figure 5) given by the difference of two Lidar surveys performed in 2003 and 2007. The authors do not specify the orientation of the computed ground deformation (subsidence? Uplift?....) nor describe the "expected" coseismic ground deformation concerning the faulting mechanism. Moreover, the map shown in Figure 5 covers approximately four years; thus, it could be affected by ground movements that are not related to the earthquake.

**Authors' reply:**

Many thanks for the comments. The authors felt very sorry that make a mistake about the DEM collected time. The DEMs before the earthquake were derived from aerial photographs shot by the Geospatial Information Authority of Japan in 1975 and 1976. Aero Asahi (2004) then used the triangulation points prepared for road construction in 1986 to orthogonalize and digitize these photos. The post-earthquake DEMs were prepared from airborne LiDAR scanning conducted by the Nakanihon Air Service on Oct 28th, 2004, the second day of the main shock and the three major aftershocks. Both sets of DEMs have a resolution of 2 m × 2 m. Since the two sets of DEMs were prepared in different ways, it is not appropriate to compare them directly in the calculation. Therefore, we used the smoothed elevations instead of the original ones for the 2004 DEMs by substituting x and y coordinates of each point into the equation of its nominal plane. The properly defined of nominal plane could also fully or partially eliminated terrain changes owing to human activities during the time gap between two sets of DEMs, since the size of the smooth window is larger than all manmade changes. In addition, in order to verify the accuracy of the calculated cosesmic ground deformation, the calculated displacements were also compared with those at points of triangulations. Totally 11 available triangulation points which buried on roads were used. The comparing results showed the difference between observed displacement and calculated displacement was small, which demonstrated the calculated cosesmic ground deformation is accurate. The detailed illustration also can be found from Konagai et al., (2009) and Zhao et al, (2012).

The orientation of the computed cosesimic ground deformation could be divided into two direction: lateral components and vertical components. Zhao et al., (2012) compared the location of earthquake-trigger landslides with the displacement field of lateral components and vertical components, respectively. The results showed that landslides clusters were found within large lateral deformation region, while the landslides seem to be off where the vertical displacement is large. So, in this study, only the lateral deformation is used. The Fig. 5 also showed the distribution of the absolute value of the lateral ground deformation. The direction of the ground deformation is

shown in Fig. R1, as only the absolute values of ground deformation were used, the direction was not considered in this analysis.

[Figure]

Fig. R1 Direction of ground deformation (a) lateral deformation (b) vertical deformation (adopted from Zhao et al., 2012)

Actually, the calculated ground deformation also could reflect the corresponding faulting mechanism. In the calculation process of cosesimic ground deformation, a cut-off window could be selected according to the regional geology to ease the tectonic displacement calculation. Zhao et al., (2012) explored the spatial distribution of lateral deformation and vertical deformation near Kajigane syncline, Japan (Fig. R1). The

results showed a 1.5-2 km band of large eastward vectors are pointing to the Kajigane syncline from the west side whereas the vectors on the east side of Kajigane syncline decreased abruptly to 0.1 m. At the same time, the vertical deformation on the west side of Kajigane syncline is upward, while the vertical deformation changed to downward on the east side of Kajigane sysncline (Fig. R1). Both the changing trend of lateral deformation and vertical deformation indicated there was a hidden reverse fault beneath the Kajigane syncline as the west side was the hanging wall.

*Reference*
*1. Aero, Asahi., 2004. Preparation of 3D spatial data in the area hit by 2004 Mid Niigata Earthquake. (Unpublished report)*
*2. Zhao, Y., Konagai, K. and Fujita. T., 2012. Multi-scale Decomposition of Co-seismic Deformation from High Resolution DEMs: a Case Study of the 2004 Mid-Niigata Earthquake. Acta Geologica Sinica(English Edition), 86(4): 1013-1021.*
*3. Konagai, K., Fujita, T., Ikeda, T. and Takatsu, S., 2009. Tectonic deformation buildup in folded mountain terrains in the October 23, 2004, Mid-Niigata earthquake. Soil Dynamics and Earthquake Engineering, 29: 261–267*

5. It is widely acknowledged that the most critical factors that affect earthquake-triggered landslides are the lithology, the slope, and the PGA. Regarding the latter, slope stability is affected not only by the PGA of the mainshock but also by the PGA of the several aftershocks, which always follow the main event. This aspect is neither introduced not discussed in the analysis.

**Authors' reply:**

Many thanks for your comments. The authors fully agreed with reviewer that slope stability is affected not only by the PGA of the mainshock but also by the PGA of the several aftershocks. Unfortunately, due to the limitation of data, the authors only have the PGA map of mainshock when preparing the manuscript. So, it may be difficult to explore the effects of aftershocks on the landslides susceptibility. This is a main limitation of this study and should be conducted. The authors will also discuss this limitation in the revised manuscript.

However, in other words, it should be noted that too many same types of influencing

factors will also lead the overfitting problems. So, at present, almost all studies only considered the PGA map of mainshock as influencing factors and neglected the effect of aftershock on the assessment of landslides susceptibility (Cao et al., 2019; Sangeeta et al., 2020; Bai et al., 2012; Tian et al., 2019; Xu and Xu, 2013; Xu et al., 2013; Li et al., 2013; Umar et al., 2014; Xu et al., 2012) and the concluded landslide susceptibility mappings were also reliable. Even so, the comments of reviewer are very useful and should be explored in further works.

*Reference*

1. *Cao, J., Zhang, Z., Wang, C.Z., Liu, J.F., Zhang, L.L., 2019. Susceptibility assessment of landslides triggered by earthquakes in the Western Sichuan Plateau. Catena, 175, 63-76.*

2. *Sangeeta, Maheshwari, B.K., Kanungo, D.P. GIS-based pre- and post-earthquake landslide susceptibility zonation with reference to 1999 Chamoli earthquake. Journal of Earth System Science 129, 55.*

3. *Bai, S.B., Wang, J., Zhang, Z.G., Cheng, C., 2012. Combined landslide susceptibility mapping after Wenchuan earthquake at the Zhouqu segment in the Bailongjiang Basin, China. Catena, 99, 18-25.*

4. *Tian, Y.Y., Xu, C., Hong, H.Y., Zhou, Q., Wang, D., 2019. Mapping earthquake-triggered landslide susceptibility by use of artificial neural network (ANN) models: an example of the 2013 Minxian (China) Mw 5.9 event. Geomatics, Natural Hazards and Risk, 10,1-25.*

5. *Xu, C., Xu, X., 2013. Controlling parameter analyses and hazard mapping for earthquake-triggered landslides: an example from a square region in Beichuan County, Sichuan Province, China. Arabian Journal of Geosciences 6, 3827–3839.*

6. *Xu, C., Xu, X., Yu, G., 2013. Landslides triggered by slipping-fault-generated earthquake on a plateau: an example of the 14 April 2010, Ms 7.1, Yushu, China earthquake. Landslides 10, 421–431*

7. *Li, W.L., Huang, R.Q., Tang, C., Xu, Q., Westen, C.V., 2013. Co-seismic Landslide Inventory and Susceptibility Mapping in the 2008 Wenchuan Earthquake Disaster Area, China. Journal of Mountain Science. 10(3), 339-354.*

8. *Umar, Z., Pradhan, B., Ahmad, A., Jebur, M.N., Tehrany, M.S., 2014. Earthquake induced landslide susceptibility mapping using an integrated ensemble frequency ratio and logistic regression models in West Sumatera Province, Indonesia. Catena, 118, 124-135.*

9. *Xu, C., Xu, X., Dai, F. and Saraf, A.K., 2012. Comparison of different models for susceptibility mapping of earthquake triggered landslides related with the 2008 Wenchuan earthquake in China. Computers & Geosciences, 46(3): 317-329.*

---

## Author Comment (AC2) · 17 May 2020

**Response to Review Comments**

Title: **The assessment of earthquake-triggered landslides susceptibility with considering coseismic ground deformation**

First of all, the authors are grateful to the reviewer, who offered many constructive suggestions to enhance the manuscript. With this reply we hope to provide detail answers to the comments of the reviewers. This is done in a point-by-point fashion below.

**Responses to the Comments Raised by Reviewer #2**

1. In this manuscript, authors present the results of statistical analyses done to the distribution of landslides induced by the Mid-Niigata earthquake (2004), Mw 6.8. Three different statistical methods (logistic regression, Artificial Neural Network and Support Vector Machine) are applied to landslide inventory at two different scales: regional and near field. In this last case, coseismic ground deformation is considered as an influencing factor in the susceptibility analysis. From the analyses, the ANN method gives the best results. The objective of the paper is to analyze the importance of the coseismic ground deformation to explain landslide distribution and the benefits of using it when preparing susceptibility maps.

The paper is properly organized and most of figures and tables are of interest.

**Authors' reply:**

Many thanks for your positive comments and valuable time to improve the manuscript.

*2.* Regarding the main objective of the paper, I miss a reflection by the authors about the true usefulness of the parameter in question in the preparation of susceptibility maps. As the authors point out in the Introduction, these maps constitute the main tool that

our society has to establish the areas prone to suffer seismic-induced landslides, and thus define an appropriate use (or restrict their occupation) of the territory. However, the parameter that constitutes the center of the article, the coseismic ground deformation, is a parameter that can only be evaluated afterwards, that is, once the earthquake has occurred. So what real use does it have? Personally, I see this parameter, as well as the distance to the surface of rupture, useful for subsequent studies, to explain why instabilities have occurred in certain contexts or areas, but not to predict their occurrence. In fact, the difference in AUC when considering/not considering this parameter is less than 5%.

**Authors' reply:**

Many thanks for your useful comments. The landslide susceptibility is defined as the likelihood of a landslide occurring in an area on the basis of the local terrain and environmental conditions (Reichenbach et al., (2018). Susceptibility measures the degree to which a local terrain can affect the **future slope movements.** In other words, it is an estimate of "where" landslides are **likely to occur.** So, the main objective of landslide susceptibility in this study is to provide planners or decision makers with the foreknowledge of landslides regions and reducing the effects of landslides which trigger by **future earthquake**. The foundation of landslides susceptibility is based on the assumption that future landslides would be more likely to occur under similar conditions to those of the existed landslides. It means the influencing factors should be derived afterwards the earthquake occurrence (landslides occurrence), especially like the influencing factors concerning the seismology. As the seismic influencing factors only could be obtained after the earthquake. For example, the PGA is the maximum change of ground shaking velocity recorded by seismometers during an earthquake and it has been commonly used in the assessment of landslide susceptibility ((Cao et al., 2019; Sangeeta et al., 2020; Bai et al., 2012; Tian et al., 2019; Xu and Xu, 2013; Xu et al., 2013; Li et al., 2013; Umar et al., 2014; Xu et al., 2012). The coseismic ground deformation is also a kind of seismic influencing factors.

In addition, the coseismic ground deformation will help to reveal the hidden subsurface damage. It should be noted that not all deformation will direct lead the landslides.

However, the area with large coseismic surface deformation often indicates that the movement of the rock mass may be further developed and the integrity of rock mass is reduced, which renders slopes prone to landslip in future earthquakes again. Zhao et al., (2012) explored the localized coseismic deformation in Kizawa (a small village), Japan after the earthquake. The results showed the calculated coseismic deformation in Kizawa is relatively larger, but the landslides are sparse. However, after a detail investigation, it found that the underground structures such as tunnels and wells were severely damaged. The road alignment of the Kizawa tunnel, which was buried 30 m beneath the ground surface, was shifted sideways 1-1.5 m to east-to-southeast direction. Furthermore, two irrigation well were dislocated at 30 m and 20 m, beneath the ground, respectively. Therefore, it is highly possible that the ground underwent some subsurface damage at locations where the large coseismic deformation. Although the deformation did not form the landslides at these locations in the 2004 Mid-Niigata earthquake, as there were accumulated deformation within the rock and soil, the landslide will easily occur in the next earthquake event. According to the comments above, the coseismic ground deformation should be regarded as a useful influencing factor in the assessment of landslides susceptibility.

Then, in order to evaluate the effects of the coseismic ground deformation on the assessment of landslides susceptibility, the Analysis of Variance method (ANOVA) has been utilized to evaluate the predictive capability of used conditional factors. The factors with higher variance values indicate a higher contribution to landslide models and vice versa. The predictive capability of eight landslide affecting factors was shown in Table 1.

Table 1. Predictive importance of different influencing factors

| Number | Influencing factor | Predictive importance |
|--------|--------------------|-----------------------|
| 1 | Lithology | 0.213 |
| 2 | Slope | 0.207 |
| 3 | PGA | 0.169 |
| 4 | Curvature | 0.125 |
| 5 | Coseismic ground deformation | 0.093 |

| | | |
|---|---|---|
| 6 | Elevation | 0.086 |
| 7 | Slope aspect | 0.057 |
| 8 | Distance to roads | 0.048 |

From Table1, it could be found the coseismic ground deformation ranked fifth place among the eight factors. The importance of coseismic surface deformation is higher than the elevation, aspect and distance from the road. Reichenbach et al., (2018) critically review the statistically based landslide susceptibility assessment literature by systematically searching for and then compiling an extensive database of 565 peer-review articles from 1983 to 2016. The results showed that elevation, aspect and distance from the road are commonly chosen as influencing factors in the assessment of landslides susceptibility. It means the coseismic ground deformation should be regarded as an important factor in the assessment of landslides susceptibility.

The AUC is a commonly used indices to evaluate the model prediction performance. At present, there are no unanimous standards to assess the increment of AUC. This means it is still debated that how much increment of AUC will be regarded as significant improvement. Most studies just considered the larger value of AUC means the better performances of the model. For example, Pham et al., (2016) conducted a comparative study of five different machine learning methods for landslide susceptibility assessment. The increment of AUC value for different models was about 0.045 (0.910-0.955). Yilmaz (2010) made a comparison of landslide susceptibility mapping methods. The increment AUC value for different models was 0.019 (0.827-0.846). Pham et al., (2017a) made a comparative study of sequential minimal optimization-based support vector machines, vote feature intervals, and logistic regression in landslide susceptibility assessment. The increment of AUC value for different models was 0.044 (0.812-0.856). Pham et al., (2017b) used the hybrid integration of multilayer perceptron neural networks and machine learning ensembles for landslide susceptibility assessment. The increment of AUC value for different models was 0.01 (0.876-0.886). Aghdam et al., 2017 conducted the landslide susceptibility assessment using a novel hybrid model of statistical bivariate methods (FR and WOE) and adaptive neuro-fuzzy inference system

(ANFIS). The increment of AUC value for different models was 0.03 (0.82-0.85). Tsangaratos and Ilia (2016) conducted the landslide susceptibility mapping using the certainty factor method, the Iterative Dichotomizer version 3 algorithm, the J48 algorithm and the modified Iterative Dichotomizer version 3 model. The validation results showed that AUC values for these models varied from 0.7766 to 0.8035. Xu et al., 2012 made a comparison of different models for susceptibility mapping of earthquake triggered landslides related with the 2008 Wenchuan earthquake in China. The results showed that the AUC values for the models varied from 0.7253 to 0.801. So, comparing the increment of AUC values in this study with above mentioned similar studies, it may be concluded that the increasing of AUC is relatively significant.

*Reference*
1. *Pham, B.T., Pradhan, B., Bui, D.T., Prakash, I., Dholakia, M.B. 2016. A comparative study of different machine learning methods for landslide susceptibility assessment: A case study of Uttarakhand area (India). Environmental Modelling & Software. 84, 240-250*
2. *Yilmaz, I., 2010. Comparison of landslide susceptibility mapping methodologies for Koyulhisar, Turkey: conditional probability, logistic regression, artificial neural networks, and support vector machine. Environmental Earth Sciences, 61(4): 821-836.*
3. *Pham, B.T., Bui, D.T., Prakash, I., Long, H.N. and Dholakia, M.B., 2017a. A comparative study of sequential minimal optimization-based support vector machines, vote feature intervals, and logistic regression in landslide susceptibility assessment using GIS. Environmental Earth Sciences, 76(10): 371.*
4. *Pham, B.T., Bui, D.T., Prakash, I., Prakash, I. and Dholakia, M.B., 2017b. Hybrid integration of Multilayer Perceptron Neural Networks and machine learning ensembles for landslide susceptibility assessment at Himalayan area (India) using GIS. Catena 149, 52–63*
5. *Aghdam, I.N., Pradhan, B., Panahi, M., 2017. Landslide susceptibility assessment using a novel hybrid model of statistical bivariate methods (FR and WOE) and adaptive neuro-fuzzy inference system (ANFIS) at southern Zagros Mountains in Iran. Environmental Earth Sciences, 76: 237.*
6. *Tsangaratos, P., Ilia, L., 2016. Landslide susceptibility mapping using a modified decision tree classifier in the Xanthi Perfection, Greece. Landslides, 13:305–320.*
7. *Xu, C., Xu, X., Dai, F. and Saraf, A.K., 2012. Comparison of different models for susceptibility mapping of earthquake triggered landslides related with the 2008 Wenchuan earthquake in China. Computers & Geosciences, 46(3): 317-329.*
8. *Zhao, Y., Konagai, K. and Fujita. T., 2012. Multi-scale Decomposition of Co-seismic Deformation*

*from High Resolution DEMs: a Case Study of the 2004 Mid-Niigata Earthquake. Acta Geologica Sinica(English Edition), 86(4): 1013-1021.*

*9. Reichenbach, P., Rossi, M., Malamud, D.B., Mihir, M. and Guzzetti, F., 2018. A review of statistically-based landslide susceptibility models. Earth-Scienc Reviews, 180: 60-91.*

*10. Cao, J., Zhang, Z., Wang, C.Z., Liu, J.F., Zhang, L.L., 2019. Susceptibility assessment of landslides triggered by earthquakes in the Western Sichuan Plateau. Catena, 175, 63-76.*

*11. Sangeeta, Maheshwari, B.K., Kanungo, D.P. GIS-based pre- and post-earthquake landslide susceptibility zonation with reference to 1999 Chamoli earthquake. Journal of Earth System Science 129, 55.*

*12. Bai, S.B., Wang, J., Zhang, Z.G., Cheng, C., 2012. Combined landslide susceptibility mapping after Wenchuan earthquake at the Zhouqu segment in the Bailongjiang Basin, China. Catena, 99, 18-25.*

*13. Tian, Y.Y., Xu, C., Hong, H.Y., Zhou, Q., Wang, D., 2019. Mapping earthquake-triggered landslide susceptibility by use of artificial neural network (ANN) models: an example of the 2013 Minxian (China) Mw 5.9 event. Geomatics, Natural Hazards and Risk, 10,1-25.*

*14. Xu, C., Xu, X., 2013. Controlling parameter analyses and hazard mapping for earthquake-triggered landslides: an example from a square region in Beichuan County, Sichuan Province, China. Arabian Journal of Geosciences 6, 3827–3839.*

*15. Xu, C., Xu, X., Yu, G., 2013. Landslides triggered by slipping-fault-generated earthquake on a plateau: an example of the 14 April 2010, Ms 7.1, Yushu, China earthquake. Landslides 10, 421–431*

*16. Li, W.L., Huang, R.Q., Tang, C., Xu, Q., Westen, C.V., 2013. Co-seismic Landslide Inventory and Susceptibility Mapping in the 2008 Wenchuan Earthquake Disaster Area, China. Journal of Mountain Science. 10(3), 339-354.*

*17. Umar, Z., Pradhan, B., Ahmad, A., Jebur, M.N., Tehrany, M.S., 2014. Earthquake induced landslide susceptibility mapping using an integrated ensemble frequency ratio and logistic regression models in West Sumatera Province, Indonesia. Catena, 118, 124-135.*

*18. Xu, C., Xu, X., Dai, F. and Saraf, A.K., 2012. Comparison of different models for susceptibility mapping of earthquake triggered landslides related with the 2008 Wenchuan earthquake in China. Computers & Geosciences, 46(3): 317-329.*